# Effects of the Phenethylamine 2-Cl-4,5-MDMA and the Synthetic Cathinone 3,4-MDPHP in Adolescent Rats: Focus on Sex Differences

**DOI:** 10.3390/biomedicines10102336

**Published:** 2022-09-20

**Authors:** Augusta Pisanu, Giacomo Lo Russo, Giuseppe Talani, Jessica Bratzu, Carlotta Siddi, Fabrizio Sanna, Marco Diana, Patrizia Porcu, Maria Antonietta De Luca, Liana Fattore

**Affiliations:** 1CNR Institute of Neuroscience-Cagliari, National Research Council-Italy, Cittadella Universitaria, Monserrato, 09042 Cagliari, Italy; 2Department of Biomedical Sciences, University of Cagliari, Cittadella Universitaria, Monserrato, 09042 Cagliari, Italy; 3‘G. Minardi’ Cognitive Neuroscience Laboratory, Department of Chemical, Physical, Mathematical and Biological Sciences, University of Sassari, Via Vienna 2, 07100 Sassari, Italy

**Keywords:** novel psychoactive substances (NPSs), cathinones, phenethylamines, abuse liability, sex differences, VTA, dopamine, corticosterone

## Abstract

The illicit drug market of novel psychoactive substances (NPSs) is expanding, becoming an alarming threat due to increasing intoxication cases and insufficient (if any) knowledge of their effects. Phenethylamine 2-chloro-4,5-methylenedioxymethamphetamine (2-Cl-4,5-MDMA) and synthetic cathinone 3,4-methylenedioxy-α-pyrrolidinohexanophenone (3,4-MDPHP) are new, emerging NPSs suggested to be particularly dangerous. This study verified whether these two new drugs (i) possess abuse liability, (ii) alter plasma corticosterone levels, and (iii) interfere with dopaminergic transmission; male and female adolescent rats were included to evaluate potential sex differences in the drug-induced effects. Findings show that the two NPSs are not able to sustain reliable self-administration behavior in rats, with cumulatively earned injections of drugs being not significantly different from cumulatively earned injections of saline in control groups. Yet, at the end of the self-administration training, females (but not males) exhibited higher plasma corticosterone levels after chronic exposure to low levels of 3,4-MDPHP (but not of 2-Cl-4,5-MDMA). Finally, electrophysiological patch-clamp recordings in the rostral ventral tegmental area (rVTA) showed that both drugs are able to increase the firing rate of rVTA dopaminergic neurons in males but not in females, confirming the sex dimorphic effects of these two NPSs. Altogether, this study demonstrates that 3,4-MDPHP and 2-Cl-4,5-MDMA are unlikely to induce dependence in occasional users but can induce other effects at both central and peripheral levels that may significantly differ between males and females.

## 1. Introduction

According to the Early Warning Advisory (EWA) from the United Nations Office on Drugs and Crime (UNODC), 1047 new psychoactive substances (NPSs) were detected in 126 countries and territories, confirming that the NPS market is expanding worldwide at an alarming rate [1]. At the end of 2020, 830 NPSs were subjected to monitoring, and 46 of these were reported for the first time in Europe [2]. In the time interval 2015–2021, 68 NPSs were placed under international control, and synthetic cannabinoids were the most common, followed by synthetic opioids and stimulants [3]. Phenethylamines and synthetic cathinones are also very common, as confirmed by recent wastewater studies [4,5]. These synthetic psychoactive substances may have abuse potential that may lead to dependence and induce toxic effects of unpredictable severity [6,7]. Recreational use of these substances is expanding among young and adult people, an effect favored by the overwhelming availability on the Internet and social media [8], and is responsible for numerous acute intoxications that may have fatal consequences [9,10]. Knowledge of the pharmacological effects of new emerging drugs is, therefore, essential to managing the clinical symptoms of intoxicated patients presenting at emergency centers. Notably, differences started being reported in the use [11] and effects [12,13] of NPSs in males and females, with women using NPSs in association with illicit and/or over-the-counter drugs more than men [14]. Preclinical studies confirmed significant sex-dependent differences in the action of novel cathinones [15] and phenethylamines [16] on the brain, in line with the vast literature reporting important sex (in animals) and gender (in humans) differences in drug addiction [17] and in other addictive behaviors [18].

Among the emerging NPSs, phenethylamine 2-chloro-4,5-methylenedioxymethamphetamine (2-Cl-4,5-MDMA, also known as 6-Cl-MDMA) and synthetic cathinone 3′,4′-methylenedioxy-α-pyrrolidinohexanophenone (3,4-MDPHP) (Figure 1) have been suggested to be particularly dangerous. The affinity and potency of these compounds, including Kd and EC50 values for main receptors, as well as their pharmacological and toxicological properties, are not known yet. Nevertheless, a few years after its first identification [19], a 29-year-old polydrug male user presented intoxicated at the hospital, where the emergency staff registered unconsciousness, hypoxia, bradycardia, and hypoventilation, while toxicological screening detected, among other drugs, a chlorinated analog of MDMA thought to be 2-Cl-4,5-MDMA [20]. It has been hypothesized that this chlorinated MDMA analog is an impurity generated during the synthesis of MDMA produced in illicit laboratories [21]. Nevertheless, this NPS is circulating on the drug market, and its central and peripheral effects are still unknown. Similar to phenethylamine 2-Cl-4,5-MDMA, another NPS lacking pharmacological characterization is synthetic cathinone 3,4-MDPHP, an α-pyrrolidinophenone structurally similar to 3′,4′-metilendiossipirovalerone (MDPV) that was identified for the first time in 2018 [22]. Aggressive behavior, delayed physical response, loss of consciousness, and coma have recently been reported in a group of nine patients testing positive for 3,4-MDPHP, although the fact that this NPS was abused concomitantly with other drugs made the recognition of its toxicological/pharmacological effects particularly difficult [23]. In another study that examined 411 intoxicated subjects presenting agitation, delirium, hallucinations, excessive motor activity, seizures, tachycardia, hypertension, and/or hyperthermia, 3,4-MDPHP was one of the NPSs most frequently found in blood and urine samples [24]. To increase the potential health risk of 3,4-MDPHP is its ability to cross the maternal-fetal blood barrier and accumulate in the fetal blood while inducing psychomotor agitation, anxiety, and mumbled speech in pregnant women [25]. Very recently, the first forensic toxicology case of death by 3,4-MDPHP without coingestion of other substances has been reported in Italy [26], further confirming that consumption of this drug puts users’ health at great risk.

An in vitro study using the dopaminergic-differentiated neuroblastoma cell line SH-SY5Y has shown that both 2-Cl-4,5-MDMA and 3,4-MDPHP strongly reduce cell viability, leading to a significant increase in reactive oxygen species (ROS), and are able to induce an increase in cellular apoptosis and in the percentage of necrotic cells, respectively [27]. Synthetic cathinones and phenethylamines include dopamine (DAT), serotonin (SERT), and norepinephrine (NET) transporters among their molecular targets and may possess abuse potential due to their ability to stimulate dopamine transmission in limbic areas [28,29,30,31,32,33]. No information is currently available on the pharmacology of 2-Cl-4,5-MDMA and 3,4-MDPHP, leaving clinicians without indication on how to manage care-seeking patients at emergency centers. Whether 2-Cl-4,5-MDMA and 3,4-MDPHP possess abuse potential (i.e., exerting reinforcing and rewarding properties) and/or are able to interfere with dopaminergic transmission is still to be determined. This study was thus undertaken to assess the abuse potential of these two compounds by evaluating their ability to sustain self-administration behaviors in adolescent rats and to alter the firing of dopaminergic neurons located in the ventral tegmental area (VTA). In light of the increasing evidence for sex-dependent differences in NPSs-induced effects, this study involved both male and female animals. Moreover, since a link between the reinforcing effects of NPSs and plasma levels of corticosterone has been reported [34], at the end of the self-administration training, plasma corticosterone levels were also measured in both sexes to assess potential drug-induced, sex-dependent hormonal stress responses.

## 2. Materials and Methods

### 2.1. Animals

A total of 42 male and 42 female Sprague Dawley rats (Charles River Laboratories, Calco, Italy) aged 21–27 days on arrival (32 for behavior and hormonal measurement + 10 for electrophysiology, for each sex) were housed 4 per cage under an inverted light–dark cycle (light on 07:00 p.m.), with standard temperature (22 ± 1 °C) and humidity (60%) conditions, and with *ad libitum* access to food and water. All animals were handled once daily for 5 min for 7 consecutive days before being assigned to either behavioral training or electrophysiological recordings. After surgery for the implantation of an intravenous (iv) catheter, animals used for the self-administration study were housed individually to avoid damage to the external component of the catheter assembly. All possible efforts were made to minimize animal pain and discomfort and to reduce the number of animals used. All experimental procedures were carried out in an animal facility according to the European Council directives (609/86 and 63/2010) and the animal policies issued by the Italian Ministry of Health (D.L. 26/2014) and were approved by the Organism for Animal Welfare (OPBA) of the University of Cagliari and the Ministry of Health, Italy.

### 2.2. Drugs

For the self-administration study, 2-chloro-4,5-methylenedioxymethamphetamine (2-Cl-4,5-MDMA, 0.1 and 0.5 mg/kg/bolus) and 3′,4′-methylenedioxy-alpha-pyrrolidinoexanophenone (3,4-MDPHP, 0.02 and 0.04 mg/kg/bolus) (Cayman Chemicals, Ann Arbor, Michigan 48108 USA) were dissolved in sterile saline (0.9%) and administered intravenously (iv) in a volume of 48 µL per infusion. All antibiotics and anesthetics were purchased as sterile solutions from local distributors. Drug doses were selected based on preliminary studies showing that they were able to increase the level of dopamine in the rat nucleus accumbens (Piras, Cadoni et al., in preparation). For the electrophysiological study, both substances were dissolved in DMSO (0.01% final concentration) and then prepared at the indicated concentration in artificial cerebrospinal fluid (ACSF) for slice perfusion.

### 2.3. Surgery

Animals were deeply anesthetized with isofluran (3%) and, under sterile conditions, were implanted with a permanent silastic catheter inserted into the external jugular vein and secured to the middle scapular region as previously described [35,36]. After surgery, each animal was individually recovered in its home cage with food and water freely available and, for the following 7 days, received a 0.1 mL intravenous infusion of enrofloxacin (8 mg/kg) followed by 0.2 mL of a heparinized sterile saline solution (250 UI/mL) to flush the antibiotic through the catheter. At the end of the 1-week recovery, food was restricted to 20 g per day to keep ~85% of free-feeding weight, and animals started the self-administration training the day after.

### 2.4. Self-Administration Apparatus and Procedure

Self-administration sessions started when rats were 36–42 days old, took place once daily, 5 days/week (from Monday to Friday), at the same time during the dark phase of the cycle (between 09:30 a.m. and 12:30 p.m.), and lasted 1 h. Sessions were carried out in chambers (31 × 26 × 33 cm), each housed in soundproof boxes (Coulbourn Instruments, Allentown, NJ, USA) and provided with two nose-poke holes, one defined as “active” and the other one as “inactive”, placed 2 cm from the floor on the short walls of the cage. A yellow light and a red light were placed inside the active and the inactive hole, respectively, and acted as discriminative visual stimuli. At the time of the self-administration training, the catheter was connected to a swivel system through a flexible metal spring, which was, in turn, connected to an infusion pump via a polyethylene tube. The swivel system allowed the animal to move freely in the operant cage. Only nose-poking activity in the “active” hole switched on the infusion pump to deliver an intravenous infusion of the drug solution, while nose-poking activity in the “inactive” hole did not activate the pump but was always recorded. Assignment of the active (drug-paired) and the inactive (not drug-paired) holes was counterbalanced between rats and remained constant for each subject throughout all the experiments. Self-administration schedules and data collection were controlled by Graphic State 2 software (Version 2.012, Coulbourn Instrument, PA, USA). Each self-administration session consisted of four different phases: (1) ready state, involving a 2 s activation of the pump to fill up the catheter with the drug solution, (2) drug availability state, allowing nose-poke responses, (3) infusion state (in case of active nose-poking), resulting in a 4 s drug infusion (volume of injection: 48 μL), followed by (4) 20 s time-out state, during which the house light was turned on, and nose-poke responses were recorded but had no programmed consequences. Rats were allowed to acquire 2-Cl-4,5-MDMA (0.1 and 0.5 mg/kg) or 3,4-MDPHP (0.02 and 0.04 mg/kg) self-administration under a continuous (fixed ratio, FR) schedule of reinforcement (FR-1) during 1 h daily sessions (5 days/week). After each self-administration session, catheters were flushed with a sterile saline solution containing heparin (1%) to ensure catheter patency. Due to catheter blockade or damage, three males and two females did not complete the training and were, therefore, excluded from the statistical analysis (see final *n* samples in the corresponding figure legend).

### 2.5. Plasma Corticosterone Levels

Two hours after the last self-administration session, rats were anesthetized, and blood was collected from the aorta and transferred to K3-EDTA-coated tubes. Blood was then centrifuged at 900× *g* for 15 min, and the resulting plasma was collected and frozen until assayed for corticosterone. Plasma corticosterone was quantified using an enzyme-linked immunosorbent assay (ELISA; #RE52211 IBL Corticosterone Enzyme Immunoassay Kit, TECAN Europe), as previously described [37]. The assay was performed according to the manufacturer’s instructions using a 96-well plate precoated with a polyclonal antibody against an antigenic site on the corticosterone molecule, a seven-point standard curve ready to use, and quality controls (low and high), all provided in the kit. Each sample was run in duplicate. Plasma corticosterone levels are expressed in ng/mL. Due to an insufficient amount of blood collected, one male and two females were not included (see the final *n* samples in the corresponding figure legend).

### 2.6. Preparation of Brain Slices

Brain slices were prepared as previously described [38,39]. After reaching deep anesthesia with vapors of isofluran (5%), male and female adolescent rats (aged 38–45 days) were euthanized, and the brains were quickly removed from the skull and transferred to a beaker with an ice-cold modified aCSF containing (in mM): 220 sucrose, 2 KCl, 0.2 CaCl_2_, 6 MgSO_4_, 26 NaHCO_3_, 1.3 NaH_2_PO_4_, and 10 D-glucose (pH 7.4, set by aeration with 95% O_2_ and 5% CO_2_). Horizontal brain slices containing the rostral part of the ventral tegmental area (rVTA) (thickness, 260 μm) were cut in ice-cold modified aCSF with the use of a VT1200S vibratome (Leica, Heidelberg, Germany). Slices were then transferred immediately to a nylon net submerged in standard aCSF containing (in mM): 126 NaCl, 3 KCl, 2 CaCl_2_, 1 MgCl_2_, 26 NaHCO_3_, 1.25 NaH_2_PO_4_, and 10 D-glucose (pH 7.4, set by aeration with 95% O_2_ and 5% CO_2_), 330 Osm. After incubation for at least 40 min at controlled temperature (35 °C) and a subsequent waiting for at least 1 h at room temperature, hemislices were transferred to the recording chamber and continuously perfused with standard aCSF at a constant flow rate of ~2 mL/min. For all recordings, the temperature of the bath was maintained at 33 °C. The effect of 2-Cl-4,5-MDMA and 3,4-MDPHP on rVTA firing rate was evaluated by slice perfusion of the drug for 5 min at different concentrations.

### 2.7. Electrophysiology

Patch-clamp recordings from rVTA dopaminergic neurons were performed as previously described [40]. Recording pipettes were prepared from borosilicate capillaries with an internal filament with the use of a P-97 Flaming Brown micropipette puller (Sutter Instruments, Novato, CA, USA). Resistance of the pipettes ranged from 4.5 to 6.0 MΩ when they were filled with an internal solution containing (in mM): 135 potassium gluconate, 10 MgCl_2_, 0.1 CaCl_2_, 1 EGTA, 10 Hepes-KOH (pH 7.3), and 2 ATP (disodium salt), 286 Osm. Signals were recorded with the use of an Axopatch 200-B amplifier (Axon Instruments Inc., San Jose, CA, United States), filtered at 2 kHz, and digitized at 5 kHz. The pClamp 10.7 software (Molecular Devices, Union City, CA, USA) was used in order to measure and analyze the firing rate and other membrane kinetic parameters of rVTA neurons as well as the occurrence of HCN-mediated hyperpolarization-activated (Ih) currents (see below). The cell-attached configuration was used to monitor the spontaneous firing rate under control conditions (average: males 4 ± 0.48 Hz (7/34), females 4.36 ± 0.68 Hz (7/30)) as well as during drug application. After obtaining a pipette-membrane seal with a GΩ resistance, at least 10 min were allowed prior to recording in order to have a stable and regular baseline firing rate. At the end of each recording, the whole-cell configuration was obtained to determine the presence of consistent Ih currents in order to confirm the identity of VTA DA neurons. A total of 10 animals (5 males + 5 females) were used for testing each drug.

### 2.8. Statistical Analyses

According to the 3Rs principles, we aim to minimize the number of animals used. To this scope, sample size calculations were performed to ensure adequate experimental group numbers to be used in the study. Based on prior studies using similar protocols [35,36,41], it was expected that a number of ≈10 subjects/group (≈60 total subjects) would be sufficient to detect significant differences between groups in the self-administration experiments and in the ex vivo assays for the determination of plasma corticosterone levels (two-way ANOVA, effect size f = 0.4, power (1-β) = 0.85, α = 0.05). Similarly, based on previous studies using similar protocols [e.g., 40], it was expected that a number of ≈4–5 subjects/group (≈16–20 total subjects) would be sufficient to detect significant differences between groups in the electrophysiology experiments (two-way ANOVA, effect size f = 0.8, power (1-β) = 0.85, α = 0.05). All calculations were carried out by using the software G*Power 3.1 (https://www.psychologie.hhu.de/arbeitsgruppen/allgemeine-psychologie-und-arbeitspsychologie/gpower).

Statistical analyses were performed using a commercially available statistical program (GraphPad Prism 8.0, GraphPad Software, San Diego, CA, USA).

For behavioral experiments, the number of nose-pokes on the active and inactive holes and the number of infusions earned during the 60 min session were recorded, and statistical analysis of the data was computed using two-way (with nose-pokes and sessions as factors) or three-way (with treatment, sex, and sessions as factors) analysis of variance (ANOVA). For the determination of plasma corticosterone levels, data are presented as means ± SEM, and statistical comparisons were performed by two-way ANOVA with drug and sex as factors. For electrophysiology experiments, data are reported as average of pooled data ± SEM, and the number of animals used (N) and the number of cells obtained (n) were indicated as N/n. Statistical comparisons of pooled data were performed by two-way ANOVA with drug and sex as factors.

In all analyses, ANOVA was followed by Tukey’s multiple comparisons test; a *p*-value of <0.05 was considered statistically significant.

## 3. Results

### 3.1. Intravenous Drug Self-Administration

To test abuse liability of 2-Cl-4,5-MDMA and 3,4-MDPHP, male and female adolescent rats (aged 36–42 days) underwent self-administration training for 2 weeks. As shown in Figure 2, neither 2-Cl-4,5-MDMA (panels A) nor 3,4-MDPHP (panels B) was able to induce in either sex a pattern of self-administration behavior consistent with abuse liability properties. Two-way ANOVA for repeated measures, with session and active/inactive nose-poke as factors, showed that both males (panels A_1_ and B_1_) and females (panels A_2_ and B_2_) did not discriminate between the active and inactive holes for either 2-Cl-4,5-MDMA (A_1_-Males: nose-poke F_(1,18)_ = 0.93, *p* = 0.35; session F_(13,234)_ = 0.82, *p* = 0.64; nose-poke × session interaction F_(13,234)_ = 1.08, *p* = 0.37. A_2_-Females: nose-poke F_(1,22)_ = 0.41, *p* = 0.53; session F_(13,286)_ = 1.88, *p* = 0.10; nose-poke × session interaction F_(13,286)_ = 1.69, *p* = 0.06) and 3,4-MDPHP (B_1_-Males: nose-poke F_(1,20)_ = 2.43, *p* = 0.13; session F_(13,260)_ = 0.42, *p* = 0.87; nose-poke × session interaction F_(13,260)_ = 0.24, *p* = 0.99. B_2_-Females: nose-poke F_(1,18)_ = 0.46, *p* = 0.51; session F_(13,234)_ = 0.65, *p* = 0.70; nose-poke × session interaction F_(13,234)_ = 0.24, *p* = 0.99]. As shown in panel C, no significant differences were found in the mean number of intravenous infusions of saline, 2-Cl-4,5-MDMA (0.1 and 0.5 mg/kg/inf), and 3,4-MDPHP (0.02 and 0.04 mg/kg/inf) self-administered by male and female rats during either the first (C_1_) or second (C_2_) week of self-administration training (three-way ANOVA for treatment, sex, and sessions as repeated measures: treatment F_(2,53)_ = 1.79, *p* = 0.18; sex F_(1,53)_ = 0.02, *p* = 0.88; session F_(1,53)_ = 1.63, *p* = 0.21; treatment × sex × sessions interaction F_(2,53)_ = 0.77, *p* = 0.47).

### 3.2. Plasma Corticosterone Levels

To determine whether 2-Cl-4,5-MDMA and 3,4-MDPHP had any effect on hypothalamic–pituitary–adrenal (HPA) axis activation, we measured corticosterone levels in plasma from male and female rats 2 h after the last self-administration session. Two-way ANOVA showed a significant effect of treatment (F_(2,50)_ = 12.78; *p* < 0.0001), a significant effect of sex (F_(1,50)_ = 62.07; *p* < 0.0001), and a significant treatment × sex interaction (F_(2,50)_ = 5.25; *p* = 0.009). As shown in Figure 3, a mean daily intake of 1.00 ± 0.2 mg/kg for males (*left*) and 0.8 ± 0.2 mg/kg for females (*right*) of 2-Cl-4,5-MDMA self-administered during the last seven sessions failed to significantly alter plasma corticosterone levels in both male (*blue bar*) and female (*red bar*) rats. By contrast, 3,4-MDPHP at a mean daily intake of 0.08 ± 0.01 mg/kg for males and 0.09 ± 0.02 mg/kg for females self-administered during the last seven sessions of training increased corticosterone levels in females (*orange bar*) (+72%; *p* < 0.001) but not in males (*green bar*). Interestingly, when comparing males and females self-administering the same drug, we found a marked sex difference in corticosterone plasma levels for 2-Cl-4,5-MDMA and 3,4-MDPHP. Indeed, when compared to the respective male counterpart, corticosterone plasma levels were higher in females that self-administered 2-Cl-4,5-MDMA (+110%; *p* < 0.01) and 3,4-MDPHP (+160%; *p* < 0.0001) (Figure 3). Corticosterone plasma levels were also increased in females that underwent saline self-administration training as compared to the corresponding male group, although this trend did not reach statistical significance (+81%; *p* = 0.08).

### 3.3. rVTA Dopamine Neurons Recordings

In order to quantitatively evaluate the effects of 2-Cl-4,5-MDMA or 3,4-MDPHP on the spontaneous firing activity of DA neurons in rVTA slices obtained from both males and females, spike discharge was recorded in the cell-attached configuration. Two-way ANOVA revealed that 5 min of slice perfusion with 2-Cl-4,5-MDMA at different concentrations (1–10 µM) caused no significant change in the firing rate of dopaminergic neurons when compared with control related to sex (F_(2,7)_ = 4.82; *p* = 0.06) or concentration (F_(1.306,4.570)_ = 1,97; *p* = 0.23). Nevertheless, when the effect of every single dose was compared with basal neuronal firing, a slight but significant effect was observed but only at the lowest dose tested (*p* = 0.04) in males (Figure 4, panels A–C).

Conversely, 5 min of 3,4-MDPHP application was able to increase the firing rate of rVTA dopaminergic neurons at all doses tested and in a dose-dependent manner (Figure 4, panels D–F). Two-way ANOVA revealed a significant effect of the drug concentration (F_(1.165,22.14)_ = 7,49; *p* = 0.009) only in males but not in females (F_(1,21)_ = 9,56; *p* = 0.005) and, hence, a significant drug × sex interaction (F_(2,38)_ = 4316; *p* = 0.02) (Figure 4, panels B,C and E,F). Notably, after a washout period of 5 min, the drug-induced effect was still evident.

## 4. Discussion

To verify the hypothesis that, similar to other phenethylamines and synthetic cathinones, 2-Cl-4,5-MDMA and 3,4-MDPHP induce positive, rewarding effects in rats, intravenous self-administration was used in this study as a highly predictive paradigm [42] widely used in preclinical research to assess the abuse potential of new compounds [43]. In animal models of addiction, in fact, MDMA proved to possess abuse liability [44], induce positive reinforcing effects [45], and sustain robust self-administration behavior in male and female rats [46,47]. Likewise, MDPV, i.e., the powerful stimulant structurally similar to 3,4-MDPHP, was shown to be consistently self-administered by rats at rates comparable with or higher than those observed during self-administration of methamphetamine [48,49]. In line with this observation, other substituted phenethylamines and synthetic cathinones, such as methylone, are robustly self-administered by rats [50,51,52], although failure to acquire self-administration behavior was recently reported for a newly emerging substituted phenethylamine, i.e., 2-(2,5-dimethoxy-4-methylphenyl)-2-methoxyethan-1-amine (BOD) [53]. All this evidence supported the hypothesis that 2-Cl-4,5-MDMA and 3,4-MDPHP could also be able to sustain reliable operant behavior in rats. However, contrary to our expectation, the two substances failed to induce a stable, sustained intake in male and female rats, at least within the range of doses and the schedules of reinforcement (i.e., the experimental protocols) used in this study. Rats self-administered only a very low number of intravenous infusions, rarely exceeding five active responses, and showed no discrimination between the active (drug-associated) and inactive (no drug-associated) hole. These observations suggest that these drugs may have little to no capability to stimulate operant behavior, though having found to induce sex-dependent alterations in plasma corticosterone levels (3,4-MDPHP, in females only) and in rVTA dopamine neuron activity (both 2-Cl-4,5-MDMA and 3,4-MDPHP, in males only).

Findings from plasma corticosterone levels were partially unexpected as well since no significant alterations in levels of this hormone in rats that underwent 2-Cl-4,5-MDMA self-administration were found. In contrast, clinical studies have reported increased plasma levels of cortisol in young male adults using MDMA [54,55,56]. The lack of significant 2-Cl-4,5-MDMA-induced alterations in corticosterone levels in females was equally unforeseen since MDMA was found to increase corticosterone levels in both adolescent and adult Sprague Dawley females [57]; however, differences in the chemical structure, dosage, route of administration, and time after drug exposure may differentially affect corticosterone levels. The lack of sex differences on corticosterone levels following 2-Cl-4,5-MDMA is similar to previous observations in Sprague Dawley rats following repeated MDMA administration, which did not affect anxiety and serotonin content [58]. More difficult to predict was the effect of 3,4-MDPHP on plasma corticosterone levels due to the lack of clinical studies and the availability of only one preclinical study, which reported increased plasma corticosterone levels in mice after acute exposure to the new synthetic cathinone derivative N-ethyl-pentedrone (NEPD) [34]. Yet, in our study, chronic exposure to low doses of 3,4-MDPHP increased corticosterone levels selectively in female rats, suggesting that males are less vulnerable to the effect of this synthetic cathinone on the stress hormonal milieu. The finding of higher plasma corticosterone levels in females that underwent saline self-administration was not surprising since similar sex-dependent differences in basal corticosterone levels were consistently reported in rats [59,60,61,62], with female rats displaying higher plasma corticosterone than males even during aging [63]. Given that estrogen regulates corticotropin-releasing factor gene expression [64], this mechanism has been suggested to account for the sex differences in basal corticosterone levels. Such a mechanism does not appear to be affected by 2-Cl-4,5-MDMA and 3,4-MDPHP self-administration, given that females still displayed higher plasma corticosterone levels than males following the self-administration of these drugs.

At last, our ex vivo electrophysiology study revealed a sex-specific and dose-dependent increase in the firing rate of rVTA dopaminergic neurons following increasing concentrations of the two drugs. Specifically, the enhancing effect was evident for 2-Cl-4,5-MDMA only at the lowest dose tested (i.e., 1 mg/kg), while for 3,4-MDPHP, it was detected at all the doses tested, dose-dependently. Notably, the stimulating effect was observed in males, while neither drug altered the firing of the rVTA neurons in females, suggesting that male users can be more vulnerable to the stimulating effect of these two emerging NPSs. In line with our findings, 3,4-methylenedioxymethamphetamine (MDMA) administration was recently reported to induce a higher increase in motor activity in adult male with respect to female mice and to affect tyrosine hydroxylase (TH) immunoreactivity in selected brain regions of male but not female mice [65]. Sex-dependent differences in VTA dopamine neuron activity after exposure to a psychostimulant could also be ascribed to the different sex hormonal milieu of males and females [66], which is known to also affect mRNA expression for specific dopamine and serotonin receptors in a region-specific manner [67]. Interestingly, the 3,4-MDPHP effect resulted not to wash out, suggesting that its action may be long-lasting or that a form of long-term plasticity on activated receptors may take place. Another aspect that will deserve further investigation is the distinction between neurons of the rVTA that project to the medial prefrontal cortex (mPFC) and/or to the nucleus accumbens (NAcc), as the effects of dopamine release from the VTA are target-specific and may have different behavioral consequences [68]. In this regard, the use of retrograde tracers with different colors for microinjections into the mPFC and NAcc and subsequent recordings of the firing rate from differentially stained cell bodies in VTA could help to disentangle the mechanisms underlying the observed dopamine activity in the VTA.

In conclusion, by combining behavioral analysis with hormonal measurement and electrophysiological recording, the present study provides the first pharmacological evaluation of (i) the positive reinforcing effects of 2-Cl-4,5-MDMA and 3,4-MDPHP in a rodent model, (ii) the effect of chronic exposure to low doses of 2-Cl-4,5-MDMA and 3,4-MDPHP on plasma corticosterone levels, and (iii) the effect of 2-Cl-4,5-MDMA and 3,4-MDPHP on rVTA dopaminergic neuron activity. Importantly, the use of animals of both sexes allowed the identification of sex-dependent significant in vivo and ex vivo effects induced by 2-Cl-4,5-MDMA and/or 3,4-MDPHP.

Findings show that 2-Cl-4,5-MDMA and 3,4-MDPHP do not induce and sustain self-administration behavior in rats, at least under the experimental conditions used in this study, suggesting that they are devoid of strong abuse potential and, hence, are unlikely to induce dependence in sporadic, occasional users. Yet, these two emerging NPSs can induce other effects at both central and peripheral levels that may significantly differ between males and females. Indeed, 3,4-MDPHP but not 2-Cl-4,5-MDMA can alter plasma corticosterone levels in female (but not male) users, while both drugs can stimulate VTA dopaminergic signaling in male (but not female) users.

## Figures and Tables

**Figure 1 biomedicines-10-02336-f001:**
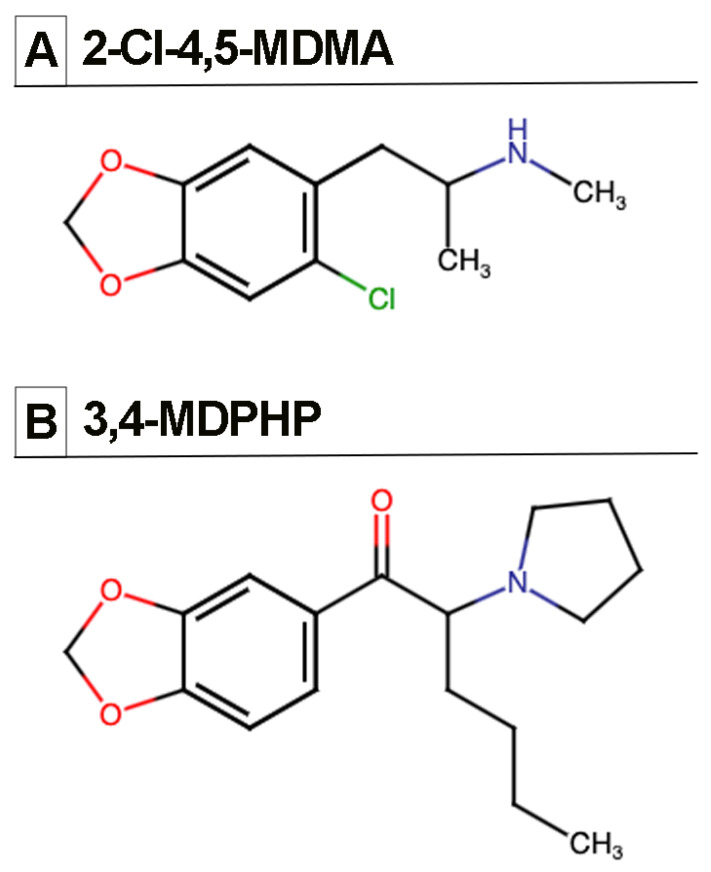
Chemical structures of the two NPSs tested in this study: (**A**) phenethylamine 2-Cl-4,5-MDMA (2-chloro-4,5-methylenedioxymethamphetamine); (**B**) synthetic cathinone 3,4-MDPHP (3′-4′-methylenedioxy-alpha-pyrrolidinoexanophenone).

**Figure 2 biomedicines-10-02336-f002:**
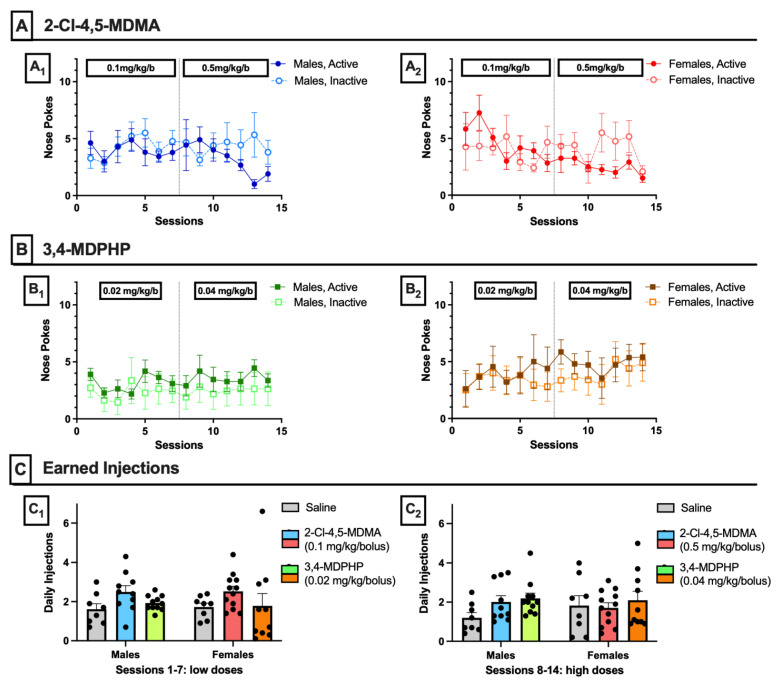
**2-Cl-4,5-MDMA and 3,4-MDPHP intravenous self-administration**. (**A**) Cumulative nose-pokes (active and inactive) performed during intravenous self-administration sessions of 2-Cl-4,5-MDMA (0.1 and 0.5 mg/kg) by male (**A_1_**) and female (**A_2_**) rats. Data are expressed as mean ± SEM of nose-pokes recorded for each group (males: *n* = 10, females: *n* = 12). (**B**) Cumulative nose-pokes (active and inactive) performed during intravenous self-administration sessions of 3,4-MDPHP (0.02 and 0.04 mg/kg) by male (**B_1_**) and female (**B_2_**) rats. Data are expressed as mean ± SEM of nose-pokes recorded for each group (males: *n* = 11, females; *n* = 10). (**C**) Average daily injections of saline, 2-Cl-4,5-MDMA, or 3,4-MDPHP earned by male and female rats during self-administration of saline and low (**C_1_**, sessions 1–7) and high doses (**C_2_**, sessions 8–14) of 2-Cl-4,5-MDMA and 3,4-MDPHP (saline: males and females *n* = 8; 2-Cl-4,5-MDMA: males *n* = 10 and females *n* = 12; 3,4-MDPHP: males *n* = 11 and females *n* = 10).

**Figure 3 biomedicines-10-02336-f003:**
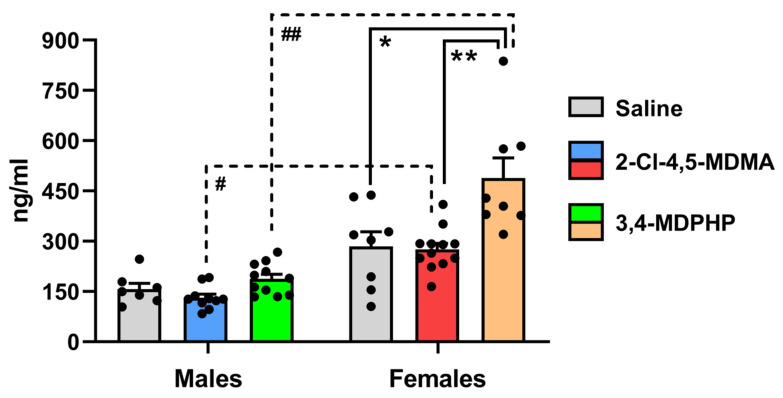
**Effect of 2-Cl-4,5-MDMA and 3,4-MDPHP administration on plasma corticosterone levels in male and female rats**. Corticosterone levels measured in plasma (expressed as nanograms per milliliter of plasma) of male and female rats at the end of the self-administration training. Data are expressed as mean ± SEM of values from 7 to 12 rats per group (males: saline, *n* = 7; 2-Cl-4,5-MDMA, *n* = 10; 3,4-MDPHP, *n* = 11; females: saline, *n* = 8; 2-Cl-4,5-MDMA, *n* = 12; 3,4-MDPHP, *n* = 8). * *p* < 0.001 and ** *p* < 0.0001 vs. the respective 3,4-MDPHP-treated group; ^#^
*p* < 0.01 and ^##^
*p* < 0.0001 vs. the respective female group; two-way ANOVA followed by Tukey’s multiple comparisons test.

**Figure 4 biomedicines-10-02336-f004:**
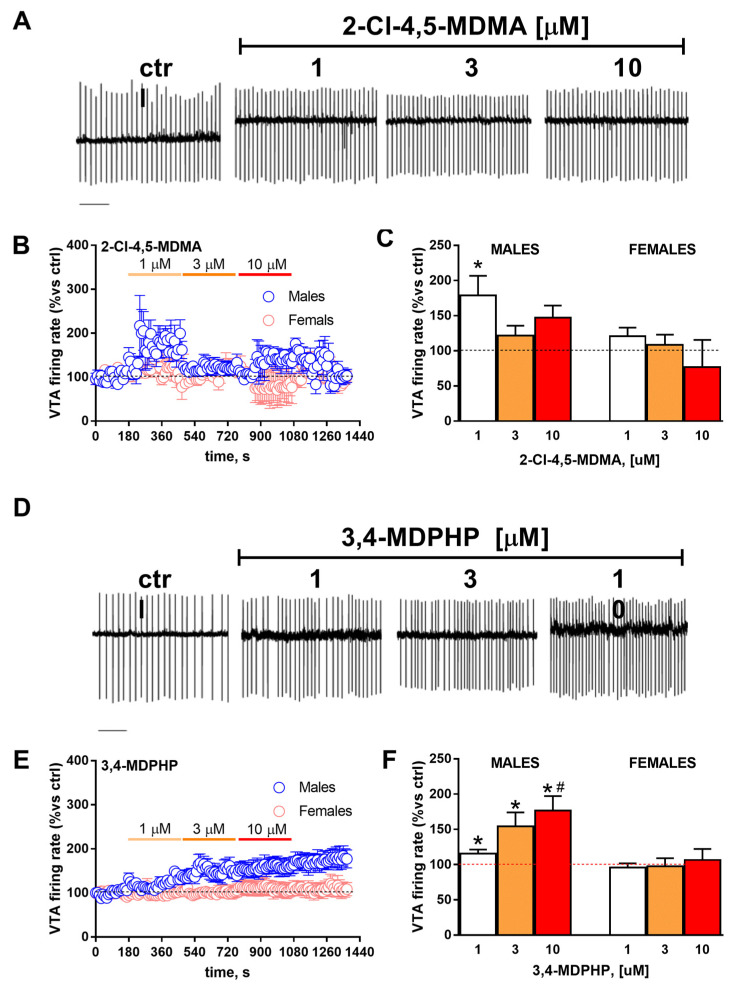
**Effect of slice perfusion with 2-Cl-4,5-MDMA or 3,4-MDPHP on firing rate of rVTA dopaminergic neurons**. (**A**) Representative traces recorded in cell-attached mode from a single dopaminergic neuron from rVTA obtained in a male rat before and after the application of different concentrations of 2-Cl-4,5-MDMA. Scale-bar, 1 s. (**B**) Scatter plot indicating the effect of bath perfusion of increasing concentrations of 2-Cl-4,5-MDMA (1–10 µM) in the firing rate of rVTA dopaminergic neurons obtained in both male and female rats (5/5). (**C**) Bar graph representing the averaged effects reported in panel B for any single concentration of 2-Cl-4,5-MDMA perfused in the slice (5/5). * *p* < 0.05 vs control values. (**D**) Representative traces recorded in cell-attached mode from a single dopaminergic neuron from rVTA obtained in a male rat before and after the application of different concentrations of 3,4-MDPHP. Scale-bar, 1 s. (**E**) Scatter plot indicating the effect of bath perfusion of increasing concentrations of 3,4-MDPHP (1–10 µM) in the firing rate of rVTA dopaminergic neurons obtained in both male and female rats (5/5). (**F**) Bar graph representing the averaged effects reported in panel E for any single concentration of 3,4-MDPHP perfused in the slice (5/5). * *p* < 0.05 vs control values, ^#^
*p* < 0.05 vs 1 µM.

## Data Availability

Not applicable.

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
