# Peer review of "Effects of the Phenethylamine 2-Cl-4,5-MDMA and the Synthetic Cathinone 3,4-MDPHP in Adolescent Rats: Focus on Sex Differences"

_biomedicines, 2022, doi:10.3390/biomedicines10102336_

Round 1

Reviewer 1 Report

The submitted paper is interesting, but in my opinion to improve the quality of the manuscript the authors should replay to several issue; in particular:

- The authors should rewrite the discussion section to give more narrative flow;

·  Regarding the animals per group, authors should specify how they decided the number and distribution. the authors have run a power analysis? Please clarify.

·      The authors should better check the manuscript for any typographical errors;

Author Response

REVIEWER #1

The submitted paper is interesting, but in my opinion to improve the quality of the manuscript the authors should replay to several issue; in particular:

- The authors should rewrite the discussion section to give more narrative flow

REPLY: The discussion has been implemented according to reviewers’ suggestions and recommendations, and has been revised to make it more narrative and fluent.

  • Regarding the animals per group, authors should specify how they decided the number and distribution. the authors have run a power analysis? Please clarify

REPLY: We are grateful to the Reviewer for raising this point. We have now formally included in the revised version of the manuscript the sample size calculations we performed before starting the experiments. Accordingly, the following has been added to the Statistics section of the manuscript (page: 6, beginning of para 2.8): “According to the 3Rs principles, we aim at minimizing the number of animals used. To this scope, sample size calculations have been performed to ensure adequate experimental groups’ numbers to be used in the study. Based on prior studies using similar protocols [e.g., 35,36,41], it is expected that a number of ≈ 10 subjects/group (≈ 60 total subjects) would be sufficient to detect significant differences between groups in the self-administration experiments and in the ex vivo assays for the determination of plasma corticosterone levels [two-way ANOVA, effect size f = 0.4, power (1-β) = 0.85, α = 0.05]. Similarly, based on previous studies using similar protocols [e.g., 40], it is expected that a number of ≈ 4-5 subjects/group (≈ 16-20 total subjects) would be sufficient to detect significant differences between groups in the electrophysiology experiments [two-way ANOVA, effect size f = 0.8, power (1-β) = 0.85, α = 0.05]. All calculations have been carried out by using the software G-Power 3.1.”

  • The authors should better check the manuscript for any typographical errors;

REPLY: the manuscript has been carefully checked for typos and other errors.

Reviewer 2 Report

Pisanu et al. reported effects of the 2-Cl-4,5-MDMA and the synthetic cathinone 3,4-MDPHP in adolescent rats. This is an interesting paper, indicating that both compounds are unlikely to induce dependence in occasional users but can induce other effects at both central and peripheral levels that may significantly differ between males and females.

There are, however, several issues to be addressed to further improve the manuscript.

1.     Although the authors showed phenomenologically the existence of gender differences, the scientific discussion of the reason why plasma corticosterone level and the VTA dopaminergic signal show differences between males and females is insufficient.

2.     The sex differences in expression level of receptors of 2-Cl-4,5-MDMA and 3,4-MDPHP such as 5-HT1A, 2A, 2B, and 2C, should be investigated in the target brain regions. Furthermore, differences in affinity of these compound, including Kd and EC50 values for receptors, should also presented.

3.     How do the authors explain the discrepancies between the effects of 2-Cl-4,5-MDMA and 3,4-MDPHP on operant behavior and rVTA dopamine neurons activity? The authors pointed out that the effects of dopamine release from the VTA are target specific and may have different behavioral consequences. Thus, the authors should examine target specificity. For example, the authors should be strongly recommended that retrograde tracers with different color are micro injected into the mPFC and NAc, respectively, then the firing rate is recorded from differentially stained cell bodies in VTA.

Author Response

REVIEWER #2

Pisanu et al. reported effects of the 2-Cl-4,5-MDMA and the synthetic cathinone 3,4-MDPHP in adolescent rats. This is an interesting paper, indicating that both compounds are unlikely to induce dependence in occasional users but can induce other effects at both central and peripheral levels that may significantly differ between males and females.

There are, however, several issues to be addressed to further improve the manuscript.

         1. Although the authors showed phenomenologically the existence of gender   differences, the scientific discussion of the reason why plasma corticosterone level and the VTA dopaminergic signal show differences between males and females is insufficient.

REPLY: According to the reviewer’s comment, in the Discussion section, the paragraph regarding sex differences in plasma corticosterone levels has been revised as follows: “Findings from plasma corticosterone levels were partially unexpected as well, since we found no significant alterations in levels of this hormone in rats that underwent 2-Cl-4,5-MDMA self-administration, in contrast with clinical studies reporting increased plasma levels of cortisol in young male adults using MDMA [54-56]. Lack of significant 2-Cl-4,5-MDMA-induced alterations in corticosterone level in females was equally unforeseen, since MDMA was found to increase corticosterone levels in both adolescent and adult Sprague-Dawley females [57]; however, differences in the chemical structure, dosage, route of administration, and time after drug exposure may differentially affect corticosterone levels. Lack of sex differences on corticosterone levels following 2-Cl-4,5-MDMA is similar to previous observations in Sprague-Dawley rats following repeated MDMA administration, which did not affect anxiety and serotonin content [58]. More difficult to predict was the effect of 3,4-MDPHP on plasma corticosterone plasma levels, due to the lack of clinical studies and the availability of only one preclinical study, which reported increased plasma corticosterone levels in mice after acute exposure to the new synthetic cathinone derivative N-ethyl-pentedrone (NEPD) [34]. Yet, in our study, chronic exposure to low doses of 3,4-MDPHP increased corticosterone levels selectively in female rats, suggesting that males are less vulnerable to the effect of this synthetic cathinones on the stress hormonal milieu. The finding of higher plasma corticosterone levels in females that underwent saline self-administration was not surprising, since similar sex-dependent differences in basal corticosterone levels were consistently reported in rats [59-62], with female rats displaying higher plasma corticosterone than males even during elderly [63]. Given that estrogen regulates corticotropin-releasing factor gene expression [64], this mechanism has been suggested to account for the sex differences in basal corticosterone levels. Such mechanism does not appear to be affected by 2-Cl-4,5-MDMA and 3,4-MDPHP self-administration, given that females still displayed higher plasma corticosterone levels than males following self-administration of these drugs.” (see pages 12 and 13).

As for the sex-differences in the VTA dopaminergic signal, we have added the following part to the Discussion (page 13): “In line with our findings, 3,4-methylenedioxymethamphetamine (MDMA) administration was recently reported to induce a higher increase of motor activity in adult male with respect to female mice and to affect tyrosine hydroxylase (TH) immunoreactivity in selected brain regions of male but not female mice [65]. Sex-dependent differences in VTA dopamine neuron activity after exposure to a psychostimulant could also be ascribed to the different sex hormonal milieu of males and females [66], which is known to also affect mRNA expression for specific dopamine and serotonin receptors in a region-specific manner [67].”

  1. The sex differences in expression level of receptors of 2-Cl-4,5-MDMA and 3,4-MDPHP such as 5-HT1A, 2A, 2B, and 2C, should be investigated in the target brain regions.

REPLY: we agree that the evaluation of sex differences in the expression level of monoamine receptors and transporters induced in the target areas by exposure to 2-Cl-4,5-MDMA and 3,4-MDPHP would be of interest. Yet, these evaluations would require a new set of experiments to obtain fresh tissue necessary to perform in situ hybridisation experiments to measure m-RNA levels of transporters and receptors. Moreover, brains of the animals that underwent self-administration training in this study have been collected after perfusion for a subsequent study aimed to evaluate neuroinflammation induced by low exposure to 2-Cl-4,5-MDMA and 3,4-MDPHP, which could be related to their reported toxicity. This follow up study will involve new batch of animals that will be exposed to these two compounds, and will allow exploring serotonin receptors besides neuroinflammation. We understand that scarcity of information about mechanisms of action of these two compounds inevitably stimulates the interest in their effects on major neurotransmitter systems, but this study was undertaken to evaluate in primis their possible abuse potential and ability to activate dopamine neurons in animals of both sexes, while follow-up studies will allow to extend the investigation to specific brain signalling pathways, including the serotoninergic one.

  1. Furthermore, differences in affinity of these compound, including Kd and EC50 values for receptors, should also presented.

REPLY: Unfortunately, affinity and potency of these compounds, including Kd and EC50 values for main receptors, are not known. We have added this information in the Introduction section – see page 2.

  1. How do the authors explain the discrepancies between the effects of 2-Cl-4,5-MDMA and 3,4-MDPHP on operant behavior and rVTA dopamine neurons activity? The authors pointed out that the effects of dopamine release from the VTA are target specific and may have different behavioral consequences. Thus, the authors should examine target specificity. For example, the authors should be strongly recommended that retrograde tracers with different color are micro injected into the mPFC and NAc, respectively, then the firing rate is recorded from differentially stained cell bodies in VTA.

REPLY: When considering discrepancies between the effects of 2-Cl-4,5-MDMA and 3,4-MDPHP on operant behavior and rVTA dopamine neurons activity it should be taken into account that animals’ behavior was under the influence of a systemic (and contingent) exposure to drugs while their effect in brain slices was evaluated by perfusing both drugs for a definite time to evaluate their ex vivo effect on rVTA dopamine neurons. Electrophysiological data thus provide indication on their action on dopamine activity, but we cannot be sure that the concentrations used in slice perfusion (1-10 µM) are comparable to those used for intravenous self-administration. It is true that most drugs that activate rVTA dopamine neurons induce positive reinforcing effects in animals and, hence, show to possess abuse potential, but it is equally true that not all drugs that activate rVTA dopamine neurons are able to sustain a stable self-administration behavior in rats.

The use of retrograde tracers with different colours for micro injections into the mPFC and NAcc would surely be useful to disentangle the mechanisms underlying drug-induced dopamine activity from the VTA and will be considered in follow-up studies. We have added this consideration in the Discussion section (page 13) as follows: “To this regard, the use of retrograde tracers with different colours for micro injections into the mPFC and NAcc and subsequent recordings of the firing rate from differentially stained cell bodies in VTA could help to disentangle the mechanisms underlying the observed dopamine activity in the VTA.”

Round 2

Reviewer 1 Report

The authors replay to all comments; in my opinion the paper is now suitable for publication

Reviewer 2 Report

The authors put an effort in revising their manuscript and addressing issues raised previously. Given the nature of the work, several aspects of the discussion remain speculative. The authors’ future study should clarify these aspects.